# Recent Progresses in Non-Dialysis Chronic Kidney Disease Patients with Hyperkalemia: Outcomes and Therapeutic Strategies

**DOI:** 10.3390/medicina59020353

**Published:** 2023-02-13

**Authors:** Ruixue Tian, Rongshan Li, Xiaoshuang Zhou

**Affiliations:** 1The Fifth Clinical Medical College of Shanxi Medical University, Taiyuan 030012, China; 2Department of Nephrology, Shanxi Provincial People’s Hospital, The Fifth Clinical Medical College of Shanxi Medical University, Shanxi Kidney Disease Institute, 29 Shuang Ta East Street, Taiyuan 030012, China

**Keywords:** hyperkalemia, chronic kidney disease, potassium

## Abstract

Chronic kidney disease (CKD) affects about 10% of the world’s population. Hyperkalemia is a life-threatening complication in patients with CKD, as it is associated with adverse cardiovascular and kidney outcomes. There are still many challenges and questions to address to improve the currently available therapeutic strategies to treat hyperkalemia, such as how to approach the emergency management of hyperkalemia. In recent years, in addition to novel oral potassium binders, great progress has been made in the application of novel kidney protective strategies, such as mineralocorticoid receptor antagonists and sodium–glucose cotransporter 2 inhibitors (SGLT2i) in hyperkalemia therapy. This review will discuss the recent advances from clinical trials in the effective management of hyperkalemia in non-dialysis CKD patients, enhancing the knowledge of physicians and internists concerning these newer agents and providing a helpful reference for clinical practice.

## 1. Introduction

Hyperkalemia is a risk factor for many life-threatening events (defined as shown in Table 1) [1]. Patients with chronic kidney disease (CKD) are more prone to develop hyperkalemia. With the progression of CKD, the renal function of patients gradually decreases, their urinary potassium excretion is further blocked, and the risk of hyperkalemia increases accordingly [2]. In addition to impaired kidney function, other factors also increase the risk of hyperkalemia, such as excessive intake of potassium-rich foods, hyperosmolar status due to high blood sugar, and the use of renin–angiotensin–aldosterone system inhibitors (RAASi) [3]. One of the first retrospective studies demonstrated that an initial serum potassium greater than 5.5 mmol/L at dialysis was strongly associated with all-cause mortality after 15 years [4]. The same conclusion was also drawn for patients on peritoneal dialysis [5].

## 2. Epidemiology of Hyperkalemia

The prevalence of hyperkalemia is highly variable, from 3% to 10% in hospitalized patients [6]. A Japanese study reported that 22.79% of non-dialysis CKD patients developed hyperkalemia (serum potassium ≥ 5.1 mmol/L) in 2016, while the prevalence of hyperkalemia in the entire cohort was only 6.79% [7]. A Denmark cohort study also found that up to 28% of CKD patients developed hyperkalemia (>5.0 mmol/L) during a 2.9-year follow-up, and 43% of patients with CKD developed hyperkalemia again during the follow-up period. A CKD prognosis consortium meta-analysis of 1.2 million patients showed that the prevalence was 17.94% when the serum potassium was greater than 5.0 mmol/L and 4.23% when the serum potassium was greater than 5.5 in the CKD cohort [8]. Furthermore, the prevalence of hyperkalemia is also influenced by the kidney function and progression of the CKD stage. In a Danish cohort of CKD stage 3a, 3b, 4, and 5 non-dialysis patients, 22.4%, 36.2%, 47.3%, and 52.2%, respectively, developed hyperkalemia during follow-up [9].

## 3. Etiology and Risk Factors

In general, risk factors for hyperkalemia include acute kidney injury (AKI), cardiovascular disease, diabetes mellitus, and a history of coronary heart disease or stroke; lower eGFR use of angiotensin-converting enzyme inhibitor (ACEi), angiotensin receptor blocker (ARB), or potassium-sparing diuretics; and non-use of thiazide or loop diuretics. The frequency of K+ monitoring should be individualized on the basis of patient comorbidities and medications, which enables proactive or preventive interventions for hyperkalemia management.

The risk factors of hyperkalemia identified in the analysis of individuals with CKD show that hyperkalemia is strongly correlated with a lower eGFR (from 15 to 105 mL/min). The incidence of hyperkalemia doubles if eGFR decreases to 15 mL/min [8]. CKD and diabetes mellitus are common risk factors for hyperkalemia because of their association with heart failure. A recent clinical observation demonstrated that hyperkalemia was higher in diabetes mellitus (20.5% vs. 12.3%). Furthermore, a lower eGFR (<60 mL/min/1.73 m^2^) predicts a higher risk of hyperkalemia in diabetes mellitus patients (25.2%) [10].

Hyperkalemia may also arise in hemodialysis patients undergoing parathyroidectomy. The possible explanation is that a decreased level of Ca+ in extracellular fluid can promote the influx of Na+ into skeletal muscle cells, which counteracts K+ efflux and Na+ influx powered by the activation of the Na/K ATPase pump [11]. It is widely accepted that morbidity and mortality benefit from mineralocorticoid receptor antagonist (MRA) therapy for heart failure. A meta-analysis of patients with heart failure or myocardial infarction revealed that the hyperkalemia rate in the MRA therapy group was 9.3% versus 4.3% in the placebo group. Hyperkalemia more often occurred in the eplerenone group than in the placebo group (5.0% vs. 2.6%), and in spironolactone trials, the use of spironolactone caused a 17.5% incidence of hyperkalemia compared with 7.5% in the placebo group [12]. The hyperkalemia frequency demonstrated a significant linear trend with an increased dose of spironolactone in end-stage renal disease patients compared with a placebo control group [13]. Moreover, the use of RAASi helps to not only improve renal and cardiovascular outcomes in CKD and heart failure patients but also increases the risk of hyperkalemia. In current clinical practice, the fear of hyperkalemia has led to the prudent use of RAASi, with submaximal doses or treatment withdrawal of RAASis causing patients with CKD or heart failure to be prone to worse outcomes [14,15]. Thus, if discontinued, RAASi therapy should be reinitiated after acute hyperkalemia has resolved.

## 4. Symptoms and Adverse Outcomes

In patients who experience serious adverse consequences, myocardial contraction disorders are typical, which in severe cases can lead to fatal arrhythmias, cardiac arrest, and even death. A retrospective trial by Brunelli et al. [16] revealed an association between patients with serum potassium ranging from 5.5 to 6.0 mmol/L and a higher risk of emergency department visits and mortality. In addition, a recently published observational trial showed that hyperkalemia patients have a higher incidence of all-cause mortality, cardiovascular events, short-term mortality, hospitalizations, and ICU admissions [17]. Bradycardia is an early sign of severe hyperkalemia that clinicians must be vigilant [18]. Hyperkalemia usually exhibits a typical peaked T wave electrocardiogram (ECG); however, the ECG sensitivity might be unreliable in patients with extreme potassium levels. A retrospective study showed that, in patients with AKI, ECG changes in peaked T waves poorly predicted the serum potassium level in both full and test samples (R^2^ = 0.03 and R^2^ = 0.01, respectively) [19]. Moreover, a large sample (124,238) of real-world clinical research about T-wave features of ECGs (amplitude and right slope) estimating the serum potassium levels concluded that there was no correlation between T-wave features and serum potassium levels [20]. Therefore, clinicians must be alert to patients with hyperkalemia who do not present with typical ECG changes in clinical practice.

As potassium is a critical participant in the contraction and conduction of the neuromuscular system, neuromuscular complications are almost universal in individuals with hyperkalemia [21]. The transmission of neural signals requires different potassium concentrations between intracellular and extracellular fluid to create an electrical and chemical charge; lacking BK potassium channels causes weakness in neuromuscular transmission [22]. Hyperkalemia shifts this critical electrical and chemical charge, which impairs function and homeostasis. Peripheral neuropathy has been shown to lead to some substantial symptoms such as pain, dysesthesia and a loss of sensation, weakness, abnormalities of gait, falls, and fall-related injury [23]. A clinical trial exploring the effects of hyperkalemia on nerve function in CKD patients showed that there was a greater decrease in the total neuropathy score of dietary-potassium-restricted patients compared with the CKD control patients [24]. Hyperkalemia can also promote membrane depolarization and the release of acetylcholine, leading to nausea, abdominal pain, and other symptoms [25]. In CKD patients, abnormal depolarization of the muscle membrane was strongly associated with the potassium level [26]. In addition to the direct harm it causes, hyperkalemia has effects on end-stage renal disease and major cardiovascular adverse events in non-dialysis CKD patients. An Italian prospective study demonstrated that non-dialysis CKD patients with hyperkalemia have a 30% higher risk of end-stage renal disease, incurring an excessive risk of mortality [27]. A recently reported study also showed that hyperkalemia patients had higher risks of premature death, hospitalization for cardiovascular and cerebrovascular adverse events, renal failure, and heart failure when compared with controls [28]. Hypokalemia could induce arrhythmia by promoting membrane hyperpolarization in a Ca^2+^-dependent manner [29]. It is worth noting that patients with hypokalemia are also at a higher risk of cardiovascular death, sudden death, and heart failure hospitalization [30]. Clinicians must recognize that serum potassium management is crucial for the prognosis of non-dialysis CKD patients, whether hyper- or hypokalemic. The harmful effects of hyperkalemia in non-dialysis CKD patients are shown in Figure 1.

## 5. Recommendations of Treatment for Hyperkalemia

### 5.1. Diet and Drug Management

To prevent hyperkaliemia, it is recommended to restrict the potassium intake to 2000–3000 mg/day. The consumption of fresh food rather than processed food, as there are more K+ additives in processed foods, represents a feasible strategy for lowering the potassium intake of CKD patients. However, dietary restrictions may not be sustainable, especially over a long time or for unpredictable periods. A cross-sectional analysis of the dietary potassium intake in a non-dialysis-dependent CKD and hemodialysis cohort found that there was no association between serum and dietary potassium or selected food groups [31]. An observational study showed that after adjusting for cardiac disease and food groups, the dietary K+ intake was not associated with all-cause mortality [32]. Based on the above considerations, whether dietary restriction of potassium intake is necessary in non-dialysis CKD patients with normal or only mild hyperkalemia needs to be further explored.

Drugs can interfere with potassium homeostasis by facilitating a transcellular potassium shift or impairing renal potassium excretion. Drugs that promote a transmembrane potassium shift are mainly represented by mannitol, beta-blockers, calcium channel blockers, and suxamethonium. Medications that paralyze renal potassium excretion mainly include ACEi, ARB, nonsteroidal anti-inflammatory drugs (NSAIDs), calcineurin inhibitors, potassium-sparing diuretics, heparin and derivatives, and aldosterone antagonists [33]. In clinical practice, a careful drug history inquiry and recognition of medications are critical as they impair potassium homeostasis, which is crucial for vulnerable patients.

### 5.2. Stabilizing the Cell Membrane

Elevated potassium destabilizes cardiac membranes, leading to cardiac arrest and sudden death, by disturbing the threshold potential of cardiac myocytes. Calcium could directly antagonize the toxic effects of hyperkalemia to restore an abnormal cell membrane potential and stabilize the myocardium; it should be administered immediately in acute hyperkalemic patients, especially with the presence of ECG changes [34]. Calcium availability formulations include calcium gluconate and calcium chloride. The European Resuscitation Council recommends the repeated use of 10 mL calcium chloride (10%) in hyperkalemic patients until the symptoms improve [35]. However, the latest guidelines recommend the use of 15–30 mL of 10% calcium gluconate solution as a priority [34]. The typical action time of calcium gluconate is approximately 3 min, and the duration is 30–60 min. The ECG of patients should be closely monitored; in cases where ECG abnormality shows no improvement or deterioration, more calcium must be given every 5–10 min [36]. When the serum K+ concentration is above 6.5 mmol/L, even if the ECG is not available, calcium solution should also be given to patients as soon as possible. The clinician must be aware that the most serious adverse effect of calcium gluconate is leakage from intravenous sites [37]. To cope with this, the infusion should be stopped and the extravasated calcium should be gently aspirated, followed by the application of cold compresses and elevation of the extremity until remission [38]. In addition, calcium carbonate may precipitate with bicarbonate-containing solutions; thus, bicarbonate solution should be avoided. Caution must be observed in heart failure patients receiving digoxin treatment as they are more prone to hyperkalemia, and calcium administration increases digoxin toxicity [39].

### 5.3. Promoting Potassium Excretion

#### 5.3.1. Intracellular Transfer

##### Insulin

Insulin is the most efficient rapid-action treatment to shift potassium; the primary shifting mechanism of insulin is by binding with glucose transporter 4 receptor on skeletal muscles, thus activating sodium–potassium adenosine triphosphatase (ATPase) to achieve potassium homeostasis control [40].

Although insulin administration to cope with hyperkalemia is typically well-tolerated, serious hypoglycemia adverse events may occur in some patients. After the administration of 10 units of insulin, hypoglycemia occurred between approximately 60 and 150 min, with a mean time of 90 min [41]. Thus, glucose should be selected in combination to reduce the risk of hypoglycemia. Studies from a medical center in the USA showed that 5 units of insulin has a similar potassium-lowering effect as 10 units but a lower incidence of hypoglycemia (9.2% vs. 19.5%) [42]. A similar conclusion was drawn in a meta-analysis showing that lower doses (5 units) of insulin could reduce the risk of hypoglycemia compared to standard doses (10 units) in patients treated for hyperkalemia without compromising potassium reduction [43]. References recommend a combination with 25–50 g of glucose when treated with 10–20 units of insulin to severely hyperkalemic patients [44,45]. Regardless of insulin doses, the serum glucose needs to be monitored closely during treatment to prevent hypoglycemia. With blood glucose concentrations greater than 200 mg/dL (11.1 mmol/L), additional glucose administration should be stopped, and insulin may be given.

##### Beta-2 Agonists

Beta-2 agonists can expedite the potassium shifts by releasing the endogenous insulin and activating the sodium–potassium ATPase of muscle cells [46]. A clinical trial in healthy individuals demonstrated that receiving 10 mg nebulized albuterol decreased the potassium level by 0.5 mmol/L, which was not observed in the placebo group [47]. Nebulization with 200 μg/mL albuterol is not only as effective as 10–15 mg glucose and 1 unit of insulin infusion but can avoid vigorous blood glucose fluctuations [48]. However, the common adverse reactions of beta-2 agonists include anxiety, tachycardia, palpitations, headache, and musculoskeletal tremors [49,50]. Albuterol administration could induce mild hyperglycemia, and a minor increase in serum lactate has also been shown [51,52].

##### Sodium Bicarbonate

In metabolic acidemia patients, sodium bicarbonate can be used to decrease blood acidity and increase urinary potassium excretion to reduce K+, although data on its efficacy are controversial. The side effects of sodium bicarbonate include hypernatremia and volume overload. Sodium bicarbonate (0.4 mmol/kg) therapy results in potassium lowering with borderline statistical significance (*p* = 0.05) [53]. A clinical trial has shown that oral sodium bicarbonate (6 g/d) for 12 weeks effectively reduces the serum potassium level (from 4.96 ± 0.62 to 4.79 ± 0.49 mmol/L) [54]. However, evidence suggests that sodium bicarbonate monotherapy might not decrease serum potassium in a timely manner when compared with insulin [42]. The addition of sodium bicarbonate therapy to intravenous insulin may also not significantly increase the efficacy of decreasing potassium [55]. Given its limited effect, for the treatment of acute hyperkalemia patients, the Kidney Disease: Improving Global Outcomes (KDIGO) do not suggest the conventional use of sodium bicarbonate [56].

#### 5.3.2. Potassium Elimination

##### Hydragogue

In patients with moderately decreased kidney function, loop diuretics and acetazolamide have been demonstrated to manage extremely severe hyperkalemia without dialysis by increasing urinary potassium losses, and the efficacy of serum potassium decreases depending on renal function. The overuse of diuretics can induce acute renal ischemic injury; thus, the guidelines suggest that they should only be used in adequate-volume resuscitated patients and not in patients with hypovolemia.

The potassium-lowering effect of loop diuretics is dose-dependent, but it might be unpredictable as it causes AKI. Acetazolamide, a carbonic-anhydrase inhibitor, increases the delivery of bicarbonate to the distal nephron, resulting in urinary potassium loss [57]. However, research shows that acetazolamide (5 mg/kg/day) for 4 weeks causes metabolic acidemia and subsequently increases the serum potassium level in Bartter syndrome patients [58]. Furthermore, patients with refractory congestive heart failure have no significant changes in K+ concentration after being treated with 125 mg/kg/day of acetazolamide [59]. Regarding the importance of the serum electrolyte balance, the safety of hydragogue treatment in non-dialysis CKD patients with hyperkalemia remains to be discussed.

##### SGLT2i

Clinical trials have demonstrated the clinical benefits of sodium-dependent glucose transporter 2 inhibitors (SGLT2i) in many type 2 diabetes patients, with different stages of CKD or heart failure, or reduced ejection fraction. Recently, analyses revealed that SGLT2i reduces the level of serum potassium.

In 2016, a clinical study reported by Yavin showed that 10 mg dapagliflozin does not affect the incidence of hyperkalemia or severe hypokalemia in patients with T2DM, whether after 24 weeks or 52 weeks [60]. However, a meta-analysis showed that SGLT2i reduced the risk of serious hyperkalemia in SGLT2i-treated T2DM patients with/without a high cardiovascular or CKD risk [61]. According to a newly reported clinical study (n = 10,142), the incidence of hyperkalemia was lower in T2DM or CKD patients who were treated with canagliflozin compared with the placebo (3.27% vs. 4.19%) [62]. Due to the potassium-lowering effect of SGLT2is only recently being revealed, there are many controversies about the effect and mechanism of SGLT2is in serum potassium regulation, which need further exploration.

##### Sodium Polystyrene Sulfonate (SPS) or Calcium Polystyrene Sulfonate (CPS)

SPS and CPS have been around for several decades as conventional therapies for hyperkalemia. SPS is a synthetic cation-bound resin, approved by the U.S. Food and Drug Administration (FDA) in the 1950s, that exchanges sodium cations for potassium in the large intestine, resulting in increased fecal potassium excretion [63]. Theoretically, the exchanging capacity of SPS is 1 mmol of potassium per 1 g. Moreover, the peak effect occurs 4–6 h after use, so SPS is not used as an emergency intervention for hyperkaliemia, even if its effect of potassium reduction is better than that of patiromer [63,64].

It is worth noting that SPS has severe but rare side effects. When SPS reaches the rectum, the role of binding potassium is the most effective. However, there can be some adverse gastrointestinal symptoms, such as colonic necrosis, constipation, and bowel obstruction. The risk of severe gastrointestinal effects significantly increased with SPS oral administration, but the risk of intestinal necrosis was not increased [65]. A higher hospitalization rate for serious gastrointestinal adverse events (intestinal ischemia/thrombosis, gastrointestinal ulceration/perforation, or resection/ostomy) was also observed in old-aged patients within 30 days of SPS use (0.2% vs. 0.04%) [66]. Gastric pneumatosis, a rare side effect induced by SPS, is drawing people’s attention [67]. In addition to the occurrence of hypocalcemia, hypernatremia, and hypocalcemia, the administration of SPS can also enhance lithium ion excretion via the gastrointestinal tract by binding and inducing low lithium [68].

The incidence of gastrointestinal symptoms may be related to the renal function of patients and the SPS dose given [69]. Combined with a large number of studies, we believe that intervention with low-dose SPS (15–30 g every day, according to the level of serum potassium) is well-tolerated and can decrease the level of potassium effectively over several weeks in CKD with hyperkalemia patients [70].

RDX7675 is the re-engineered calcium salt of polystyrene sulfonate-based resin with greater palatability than SPS. The potassium-binding capacity of RDX7675 (1.14 mEq/g) was higher than patiromer (0.63 mEq/g) or SPS (0.73 mEq/g) [71]. In general, RDX7675 and SPS were well tolerated during 4 days of continuous administration in healthy people, indicating that RDX7675 could be used for patients with hyperkalemia [72]. Patients with hyperkalemia may benefit from long-term potassium-binding agents, initiated at a lower or recommended dose and altered according to the levels of serum potassium.

##### Patiromer

Patiromer consists of smooth beads approximately 100 mm in diameter. Research in animals showed that patiromer is not absorbed in the gut and is excreted in the feces [73]. Patiromer can bind to more than 8 mmoL of potassium per 1 g under physiologic conditions.

A real-world retrospective study based on large data evaluated the action of patiromer in hyperkalemia (K+ ≥ 5.1 mmol/L) veterans with heart failure, diabetes, or CKD. The mean reduction in serum potassium was 1.0 mmol/L from the baseline for 6 months (*p* < 0.001). After patiromer was used for 3–6 months, 71% of patients had K+ < 5.1 mmol/L, which indicates that patiromer could be applied for the chronic management of hyperkalemia [74]. Spironolactone is effective for reducing blood pressure in patients with hypertension. In patients with resistant hypertension and CKD, patiromer (84 g once daily) enabled patients to safely continue spironolactone (25 mg once daily) treatment with less hyperkalemia for 12 weeks [75]. Patients with heart failure have a high risk of hyperkalemia due to the use of RAASi. A pooled analysis of three randomized trials demonstrated that patiromer was generally similarly effective in reducing the serum K+ in patients with and without heart failure [76]. Thus, clinicians must pay attention to the agents that affect potassium when using patiromer to deal with hyperkalemia so as to achieve the best effect of lowering blood potassium.

Patiromer may not be used for emergency treatment because it delays its effects. Interestingly, the results of a study about patients with end-stage renal disease with severe serum potassium (≥6.0 mmol/L) in the emergency department showed that a single dose of 25.2 g of oral patiromer reduced the serum potassium within 2 h, and safely enough [77]. However, the number of participants was limited and the duration of the study was short. According to a newly published clinical study, after a single dose of patiromer (8.4 g, 16.8 g, or 25.2 g) treatment, the mean potassium reduction was 0.50 mEq/L (*p* < 0.001) after 6 h [78]. Thus, we think that patiromer monotherapy may be a promising selection for managing patients with acute, non-life-threatening hyperkalemia. Further clinical evidence is needed for evaluating the role of patiromer in acute hyperkalemia.

Patiromer adsorbs potassium, sodium, and magnesium in the large intestine of the gut, but we did not know what effects it has on the electrolyte balance. In a study of patiromer’s effects on mineral metabolism, people were given 8.4 g of patiromer per day for 4 weeks. Patiromer decreased the phosphate in the urine of all patients (*p* = 0.004) and lowered the level of phosphate in the serum and urine in some patients with hyperphosphatemia. It also decreased parathyroid hormone (*p* < 0.001) and 1,25-dihydroxyvitamin D (*p* = 0.05) but had no impact on serum calcium (*p* = 0.78) [79]. No serious gastrointestinal adverse events were reported during the study. Thus, patiromer is safe enough and has little effect on the electrolyte balance. It may bind magnesium and lead to hypomagnesemia in some patients. Overall, many clinical trials demonstrated that patiromer is effective for the management and treatment of chronic or acute hyperkalemia. Patiromer appears to be well tolerated with long-term use and no serious adverse events.

##### Sodium Zirconium Cyclosilicate (ZS-9)

ZS-9 is a powder composed of zirconium, silicon, and oxygen, which are arranged to form cation-binding pores. In vitro, this agent was found to rapidly reduce the serum potassium, sequestering potassium nine times more than SPS, and has been shown to safely maintain normokalemia [80]. The characteristics of different potassium binders are summarized in Table 2.

Spinowitz et al. [81] reported a study in which 751 patients with hyperkalemia (≥5.1 mmol/L) received ZS-9 5 g once daily; 99% achieved normokalemia during the ≤12-month maintenance phase. A clinical trial to test the safety and efficacy showed that, in patients with CKD and hyperkalemia, i.e., the 10 g ZS-9-treated group, the serum potassium concentration decreased significantly by 0.11 mmol/L at 1 h compared to the placebo group. After 38 h, 41.7% of patients with 10 g ZS-9 therapy had a potassium reduction greater than 1.0 mmol/L compared to 3.4% in the placebo group. No serious adverse events were reported in this short-term study. Furthermore, there were no clinical changes in serum calcium, magnesium, or sodium nor in kidney function index serum creatinine and blood urea nitrogen with ZS-9 intervention [82].

There was a trial that evaluated the efficacy and safety of ZS-9 in patients with hyperkalemia. Patients received ZS-9 (5, 10, or 15 g) or a placebo for 48 h or 28 days, respectively. After 48 h, 10 g of ZS-9 three times daily caused the serum potassium to decrease from 5.6 to 4.5 mEq/L at 48 h, with 84% of patients reaching normokalemia by 24 h and 98% by 48 h. Furthermore, the proportion of patients with K+ <5.1 mEq/L during days 8–29 was 80%, 90%, and 94% in the 5 g, 10 g, and 15 g ZS-9 groups, respectively, while it was 46% in the placebo group. Patients accepting RAASi therapy achieved a mean serum potassium reduction of –1.02 mEq/L by hour 48, which was similar to all the participants. Edema was a more common adverse event in the 15 g ZS-9 group (6%) compared to 14% of patients in the placebo. Hypokalemia developed in five (10%) and six patients (11%) in the 10 g and 15 g ZS-9 groups, while none developed hypokalemia in the 5 g and placebo groups [83].

### 5.4. Dialysis

Ultimately, if drug treatment is not satisfactory, CKD patients with acute hemodialysis should consider lowering serum potassium level through dialysis. Dialysis is the most effective way to eliminate excess potassium.

The recommendations for the management of hyperkalemia are shown in Figure 2.

## 6. Conclusions

The management of serum potassium in non-dialysis CKD patients is very important for their clinical outcomes. The management methods of hyperkalemia mainly include drug therapy, diet management, etc. In terms of drug therapy, traditional oral potassium lowering drugs have been used clinically for many years, but they have serious side effects and lack of long-term clinical trial data. In recent years, new oral potassium lowering agents patiromer or ZS-9 can safely and effectively control serum potassium levels, reduce the need for monitoring, and meet the clinical needs. In terms of diet, the guidelines recommend diet and potassium restriction in CKD patients with hyperkalemia, but recent data suggest that the effect of dietary restriction on clinical outcomes in patients needs to be further studied. 

## Figures and Tables

**Figure 1 medicina-59-00353-f001:**
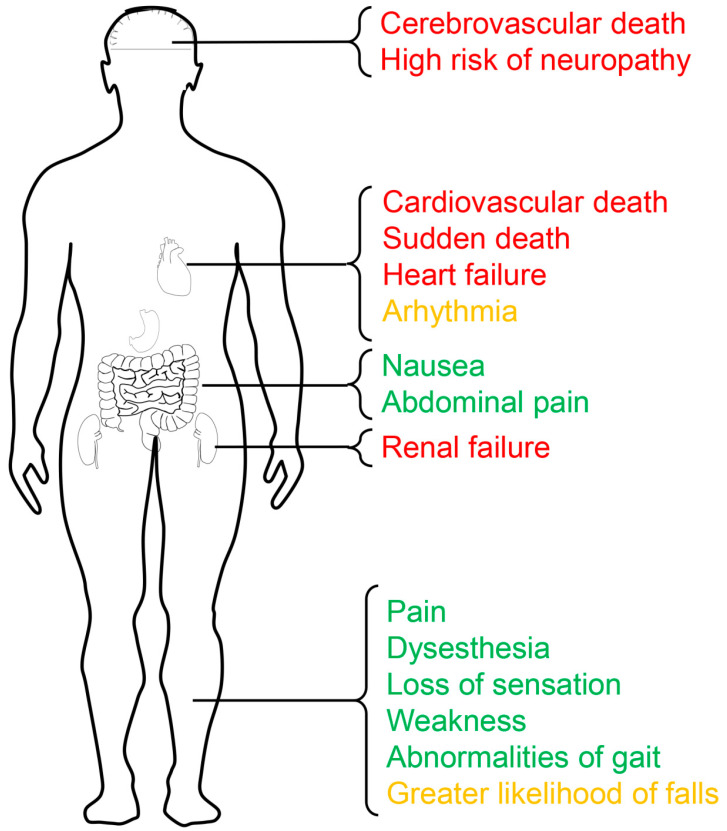
The harmful effects of hyperkalemia in non-dialysis chronic kidney patients. Cerebrovascular death, neuropathy, cardiovascular death, sudden death, heart failure, and renal failure are highlighted in red, meaning that they are very dangerous; arrhythmia and falls are highlighted in yellow, meaning that they are moderately dangerous; nausea, abdominal pain, dysesthesia, loss of sensation, weakness, and abnormalities of gait are highlighted in green, meaning that they are generally dangerous.

**Figure 2 medicina-59-00353-f002:**
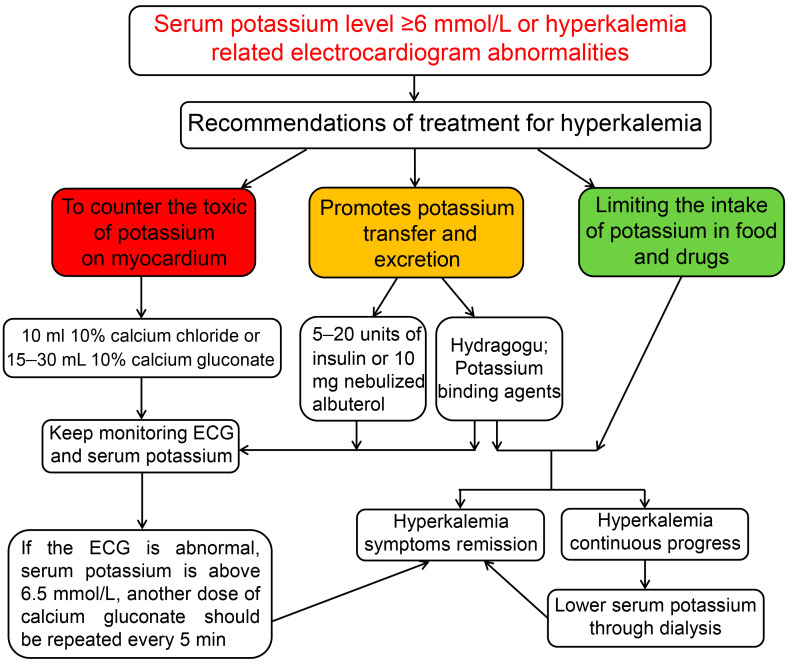
Recommendations for the management of hyperkalemia in non-dialysis chronic kidney disease patients. First, the severity of hyperkalemia is graded according to the plasma level of potassium. For mild hyperkalemia stage, restrict potassium intake to less than 2400 mg/day. For moderate hyperkalemia or severe hyperkalemia, in any hyperkalemic patients presenting with ECG changes suggesting hyperkalemia, treatment must be implemented for stabilizing the cell membrane drug; then, potassium elimination can be used according to the level of serum hyperkalemia. The ECG and serum hyperkalemia should be monitored promptly.

**Table 1 medicina-59-00353-t001:** Definition of hyperkalemia.

	Mild Hyperkalemia	Moderate Hyperkalemia	Severe Hyperkalemia
The level of potassium (mmol/L)	5.5–5.9	6.0–6.5	>6.5
ECG changes	–	–/+	+

**Table 2 medicina-59-00353-t002:** Characteristics of potassium-binding agents.

	SPS	Patiromer	ZS-9
Molecular structure	synthetic cation-bound resin	smooth spherical beads approximately 100 mm in diameter	a three-dimensional crystalline lattice structure composed of zirconium, silicon, and oxygen
Molecular mechanism	Nonspecific potassium binding in exchange for sodium	Nonspecific potassium binding in exchange for calcium	Selective potassium binding in exchange for sodium
	colon	distal colon	entire intestinal tract
Dosage	15–30 g orally/day	4.2–50.4 g/day	1.25–15 g/day
Onset of effect	1–2 h	2–6 h	1–2 h
Adverse events	gastrointestinal disorders hypernatremia hypokalemia alkalosis volume overload	gastrointestinal disorders hypokalemia hypomagnesemia	gastrointestinal disorders hypokalemia edema

## Data Availability

Not applicable.

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
