# Peer review of "Recent Progresses in Non-Dialysis Chronic Kidney Disease Patients with Hyperkalemia: Outcomes and Therapeutic Strategies"

_medicina, 2023, doi:10.3390/medicina59020353_

Round 1

Reviewer 1 Report

Review of the topic of hyperkalemia in CKD. No new information offered. If looking for a simple review of this topic, this manuscript is acceptable as is. If looking for new insights or new approaches, then this article has nothing to offer.

1. "Main question addressed by the research"  -- as stated, this manuscript is NOT new research.  IT IS A REVIEW.
2.  A good review is always relevant (therefore interesting) because it brings the total topic into focus. Readers can learn from a thorough review.
3.  Well written
4.  CONCLUSIONS: a reasonable review because it discussed known information encompassed  by the title of the article.
5. Needs a spell check.

Author Response

Thank you for your letter and for the reviewers’ comments concerning our manuscript entitled “Recent Progresses in Non-Dialysis Chronic Kidney Disease Pa-tients with Hyperkalemia: Outcomes and Therapeutic Strategies” (ID: medicina-2170918). Those comments are all valuable and very helpful for revising and improving our paper. We have studied comments carefully and have made modification which we hope meet with approval. Revised portion are marked in red in the paper. The main corrections in the paper and the responds to the reviewer’s comments are as flowing:

 Review 1

Question 1: "Main question addressed by the research" -- as stated, this manuscript is NOT new research.  IT IS A REVIEW.

Answer: Thanks for your comments, this review will to discuss the recent progresses from clinical trials in the resultful management of hyperkalemia in non-dialysis CKD patients, enhancing the knowledge of physicians and internists with respect of these newer agents, to provide a helpful reference for clinical practice.

Question 2: A good review is always relevant (therefore interesting) because it brings the total topic into focus. Readers can learn from a thorough review.

Answer: We are very thanks for your warm comments.

Question 3: Well written

Answer: Thank you for your comments.

Question 4:  CONCLUSIONS: a reasonable review because it discussed known information encompassed by the title of the article.

Answer: We are very thanks for your comments.

Question 5: Needs a spell check.

Answer: We are very sorry for the spelling mistake, the text has been spell-checked and polished by professional language editing service.

We tried our best to improve the manuscript and have made further modifications to improve the logic of the article and make it easier to read. These changes will not influence the content and framework of the paper, here we did not list the detailed changes but highlighted in red.

We appreciate for Editors/Reviewers’ warm work earnestly, and hope that the correction will meet with approval. Once again, thank you very much for your comments and suggestions.

Yours sincerely,

Xiaoshuang Zhou

Department of Nephrology, Shanxi Provincial People's Hospital; The Fifth Clinical Medical College of Shanxi Medical University; Shanxi Kidney Disease Institute, 29 Shuang Ta East Street, Taiyuan, Shanxi Province, China, 030012

Reviewer 2 Report

This is a review regarding hyperkalemia management in non-dialysis patients. It is presented as a list of possible treatments in this setting.

Language is unsatisfactory, and it requires a professional assessment. Many phrases are obscure and limit readability. There are several grammatical errors.

Other issues

1. The authors aim to present “recent progresses” in hyperkaliemia in non-dialysis patients. However, it is not clear where these progresses are in the main text. Can you please add a paragraph highlighting the novelties?

2. Figure 2. What’s the meaning of “hyperkalassium”? This figure must be carefully revised since it’s hard to see the administration of immunosuppressants without a thorough clinical assessment. It’s difficult to understand why the authors included AKI in this flowchart since it refers to hyperkaliemia.

Author Response

Thank you for your letter and for the reviewers’ comments concerning our manuscript entitled “Recent Progresses in Non-Dialysis Chronic Kidney Disease Pa-tients with Hyperkalemia: Outcomes and Therapeutic Strategies” (ID: medicina-2170918). Those comments are all valuable and very helpful for revising and improving our paper. We have studied comments carefully and have made modification which we hope meet with approval. Revised portion are marked in red in the paper. The main corrections in the paper and the responds to the reviewer’s comments are as flowing:

 Review 2

This is a review regarding hyperkalemia management in non-dialysis patients. It is presented as a list of possible treatments in this setting.

Question 1: Language is unsatisfactory, and it requires a professional assessment. Many phrases are obscure and limit readability. There are several grammatical errors.

Answer: We are very sorry for our incorrect written, text has been edited and polished by recommend language editing service.

Question 2: The authors aim to present “recent progresses” in hyperkaliemia in non-dialysis patients. However, it is not clear where these progresses are in the main text. Can you please add a paragraph highlighting the novelties?

Answer: We apologize for causing your confusion. The references we cite are almost all within five years to present “recent progresses”. Furthermore, a paragraph to show might be illogical, some content has been updated to highlight the recent progresses in the paper, they have been marked in red. We are looking forward to receiving your agreement. Thank you again for your enthusiastic work.

Question 3: Figure 2. What’s the meaning of “hyperkalassium”? This figure must be carefully revised since it’s hard to see the administration of immunosuppressants without a thorough clinical assessment. It’s difficult to understand why the authors included AKI in this flowchart since it refers to hyperkaliemia.

Answer: We are very sorry for this mistake, the figure 2 has been carefully revised according to the reviewer's reminder, and hope this modification will meet with approval, thank you very much.

We tried our best to improve the manuscript and have made further modifications to improve the logic of the article and make it easier to read. These changes will not influence the content and framework of the paper, here we did not list the detailed changes but highlighted in red.

We appreciate for Editors/Reviewers’ warm work earnestly, and hope that the correction will meet with approval. Once again, thank you very much for your comments and suggestions.

Yours sincerely,

Xiaoshuang Zhou

Department of Nephrology, Shanxi Provincial People's Hospital; The Fifth Clinical Medical College of Shanxi Medical University; Shanxi Kidney Disease Institute, 29 Shuang Ta East Street, Taiyuan, Shanxi Province, China, 030012

Feb. 5, 2023

Round 2

Reviewer 2 Report

The manuscript has been impressively improved. Language is fine. I have no other issues to raise.